# Diffuse C-Cells Hyperplasia Is the Source of False Positive Calcitonin Measurement in FNA Washout Fluids of Thyroid Nodules: A Rational Clinical Approach to Avoiding Unnecessary Surgery

**DOI:** 10.3390/cancers16010210

**Published:** 2024-01-02

**Authors:** Chiara Mura, Rossella Rodia, Silvia Corrias, Antonello Cappai, Maria Letizia Lai, Gian Luigi Canu, Fabio Medas, Pietro Giorgio Calò, Stefano Mariotti, Francesco Boi

**Affiliations:** 1Endocrinology Unit, Department of Medical Sciences and Public Health, University of Cagliari, 09100 Cagliari, Italy; chiara.mura@hotmail.com (C.M.); rossellarodia@hotmail.it (R.R.); scorrias92@gmail.com (S.C.); acappai@unica.it (A.C.); mariottistefano48@gmail.com (S.M.); 2Pathology Unit, San Giovanni di Dio Hospital, 09100 Cagliari, Italy; marialetizialai@yahoo.it; 3Surgery Unit, Department of Surgical Sciences, University of Cagliari, 09100 Cagliari, Italy; gianl.canu@unica.it (G.L.C.); fabio.medas@unica.it (F.M.); pgcalo@unica.it (P.G.C.)

**Keywords:** calcitonin, FNA-CT, CCH, MTC, thyroidectomy

## Abstract

**Simple Summary:**

The usefulness of calcitonin measurement in fine needle aspiration (FNA-CT) for the diagnosis of medullary thyroid carcinoma (MTC) has been confirmed by several authors. During the review process of records of our patients with thyroid nodules (TNs) and elevated serum CT, we focused our attention on 11 patients with TNs showing at histology thyroid neoplasms other than MTC, as compared with a MTC patients’ group. We showed that a high FNA-CT concentration is not only found in MTC but may also be observed in benign/malignant TNs coexisting with diffuse C-cells hyperplasia (CCH) due to contamination of the needle through the CCH area. Since diffuse CCH may cause false positive FNA-CT results, we propose a cautious use of FNA-CT in patients with TNs with increased serum CT and high FNA-CT values to avoid unnecessary surgery.

**Abstract:**

Purpose: The FNA-CT is useful for the diagnosis of MTC. The aim of this study was to evaluate the performance of FNA-CT in TNs coexisting with CCH. Methods: This study retrospectively reviewed the records of 11 patients with TNs submitted to thyroidectomy on the basis of elevated basal and/or stimulated serum CT values, which at histology were not confirmed to be MTC. The results obtained in this group were compared with those of a previously reported group of histologically proven MTC patients submitted to an identical presurgical evaluation. All patients, negative for known mutations in the RET proto-oncogene, were preoperatively submitted to neck ultrasound, FNA-cytology, and FNA-CT. Results: Approximately 6 of 11 patients showed increased (>36 ng/mL, as established in previous studies not involving patients with CCH) FNA-CT. All these patients showed diffuse CCH at histology in the thyroid lobe submitted to FNA; 5 of them were benign at histology, while only one was malignant (papillary thyroid carcinoma, PTC). The remaining 5 of 11 patients had low FNA-CT (<36 ng/mL), and all of them showed only focal CCH in the lobe submitted to FNA; three of them were malignant (2 PTC, 1 follicular carcinoma), while two were benign. Conclusions: Employing the currently proposed cut-off values, false-positive FNA-CT results may be observed in benign/malignant TNs with coexisting diffuse CCH. FNA-CT must therefore be cautiously used in the diagnostic approach for patients with TNs and a slightly increased basal or stimulated serum CT concentration in order to avoid unnecessary surgery.

## 1. Introduction

Early detection of medullary thyroid carcinoma (MTC) has important implications for the treatment of MTC and the patient’s outcome, improving the prognosis and the follow-up of the disease [1]. Calcitonin (CT) measurement in the washout fluid of fine needle aspiration (FNA-CT) of thyroid nodules (TNs) and/or neck masses has been largely used since 2007 [2] to identify primary and/or recurrent/metastatic MTC and is currently recommended by several consensuses and guidelines of numerous scientific societies [1,3].

The diagnostic power of FNA-CT has been recently confirmed in a recent meta-analysis [4] to be significantly higher (with a pooled sensitivity of 98% with a 95% CI of 96% to 100%) than FNA-cytology (with a pooled sensitivity of 54% with a 95% CI of 35 to 73%). On the other hand, the measurement of serum CT, which may represent a sensitive marker of both MTC and C-cell hyperplasia (CCH) [5,6], can be influenced by several factors (such as proton pump inhibitors, renal failure, and some endocrine diseases), particularly in the presence of a mild CT increase [5,7,8,9]. Because of these limitations, the routine use of serum CT in patients with TN has not yet been generally proposed by current guidelines [1,4,5]. Moreover, stimulated serum CT induced by pentagastrin (Pg) [10] and/or calcium [11], which has been widely employed in the last decades, demonstrated good diagnostic performance only for values surpassing the proposed cut-off, with a high rate of false positive results and uncertain significance for intermediate borderline values [11,12]. These interferences may lead to false-positive MTC diagnoses, resulting in an unnecessary thyroidectomy [13].

Although FNA-CT has a very high diagnostic value in detecting MTC, it should be noted that CCH concomitant to TNs may be associated with increased FNA-CT values, potentially representing another source of false positive results, as preliminary reports in some cases of patients with CCH submitted to surgery [14,15]. Since non-neoplastic, reactive, or physiologic CCH (biologically different from neoplastic CCH associated with familial MTC and MEN 2 syndromes) is frequently observed in patients with multinodular goiter (MNG) and Hashimoto’s thyroiditis [16], the rate and relevance of false positive FNA-CT, potentially caused by this condition, may be underestimated.

With these concepts in mind, in order to re-evaluate the clinical relevance of FNA-CT in the presence of non-neoplastic CCH without MTC at histology, we retrospectively reviewed records of a highly selected group of patients with TNs with high basal and/or stimulated serum CT who were submitted to FNA-CT and subsequent total thyroidectomy. We then compared it with the records of our original group of patients who presented MTC at histology [2]. The results showed that non-neoplastic CCH associated with benign or non-MTC thyroid cancer may be the source of false-positive FNA-CT results.

## 2. Materials and Methods

### 2.1. Patients

This study retrospectively evaluated the records of a study group of 11 patients seen between 2008 and 2018 at the outpatient Clinic of the Endocrinology Unit, University Hospital of Cagliari, submitted to total thyroidectomy for the presence of TN, elevated basal and/or stimulated serum CT, and preoperative FNA-CT values, in whom the presence of MTC was not confirmed at histology. The results were then compared to those of the 10 patients in our original series [2] with elevated basal and/or stimulated serum CT, high preoperative FNA-CT results, and confirmed MTC at histology.

We excluded potential causes of elevated CT that could interfere with the results: all the patients involved were non-smokers, they did not use medications known to influence CT levels, and they did not present any other known interfering factors. Three of the 11 patients in the present study group had presurgical serum-positive thyroid autoantibodies (2 anti-thyroglobulin and 3 anti-thyroperoxidase) due to the coexistence of Hashimoto’s disease, later confirmed histologically by the presence of lymphocytic thyroiditis.

The protocol of this study was reviewed and approved by the Institutional Review Board of the University Hospital of Cagliari. The ethical code of this study is: Outcome PG/2017/3273-3.42/2022. Each patient provided written informed consent after a full explanation of the purpose and nature of all procedures used. All the patients were submitted to an accurate physical examination, neck US, US-guided FNA for conventional cytology, and FNA-CT; a search for RET proto-oncogene mutations was also performed in all cases.

### 2.2. Thyroid Ultrasound

Thyroid ultrasound (US) was performed using Siemens Antares color Doppler system equipment (Siemens Medical Solutions, Issaquah, WA, USA). All TNs were identified and localized, and their diameters were measured. To evaluate US features, all TNs were classified according to EU-TIRADS criteria [17] and, when needed, submitted to FNA under US guidance.

### 2.3. Hormonal Assays

Serum CT and FNA-CT were assessed by an ultrasensitive chemiluminescent assay (Immulite 2000 Calcitonin; Diagnostic Products Corp., Los Angeles, CA, USA, distributed by Medical Systems Corp., Genoa, Italy). Normal values for serum CT were less than 18 pg/mL for males and less than 12 pg/mL for females.

Serum CT was assayed pre- and post-stimulation with calcium gluconate. For this purpose, CT was assayed before and 2, 5, and 15 min after the IV bolus of 2.5 mg/kg of calcium gluconate. Serum CT after calcium stimulation was considered normal < 30 pg/mL, borderline between 30 and 100 pg/mL, and positive > 100 pg/mL in males and >95 pg/mL in females [18,19]. All the assays were performed in the laboratory of the “Duilio Casula” University Hospital of Cagliari.

### 2.4. Cytology and FNA-CT

US-guided FNA was performed according to the current European Guidelines [20], employing 22–25-gauge needles and a 10 mL syringe [21]. Cytological examination was performed by an experienced thyroid pathologist (M.L.L.), who was unaware of FNA-CT results in order to avoid potential conditioning in cytological examination. According to criteria employed in the study [22], cytological results were classified as: not diagnostic (THY 1), non-malignant/benign (THY 2), low-risk indeterminate lesion (THY 3-A), high-risk indeterminate lesion (THY 3-B), suspicious for malignancy (THY 4), and malignant (THY 5). After smear preparation, the needle was washed out with 500 µL CT-free serum dilution buffer, and the solution was processed for FNA-CT measurement. The value of the FNA-CT cut-off considered as an expression of local CT production by C-cells was 36 pg/mL, since all FNA-CT above this cut-off resulted in the diagnosis of MTC at histology, as widely reported in our previous study not involving patients with CCH [2].

Finally, in this series, the criteria for selecting patients for total thyroidectomy were as follows: Surgery was advised for patients with a THY 4 cytology and suggested in patients with indeterminate THY 3-B cytology; it was also proposed for patients with THY 3-A and THY 2 cytology when TN were associated with high FNA-CT levels and/or in the presence of suspicious US features and large MNG. Although it is well known that the risk of thyroid cancer is low in patients with large or slowly growing thyroid nodules [23], in our patient with MNG, the decision for surgery was based on the presence of relevant compressive symptoms, as suggested by current guidelines [24,25].

### 2.5. Histology

On surgical specimens, the presence of benign TN (follicular adenoma [FA], hyperplastic nodule [HN], and MNG) or malignant thyroid lesions (papillary thyroid cancer [PTC] and follicular carcinoma [FC]) was identified by common criteria [26]. The histological diagnosis of diffuse and focal CCH was made on the basis of the number of C-cells observed in a single microscopic field using standard pathological techniques, including the search for CT expression by immunohistochemistry. C-cell hyperplasia was diagnosed when at least one area with >50 C-cells per one low-power field (×100) was identified [27]. Specifically, focal CCH was defined as focal when segmental proliferation of C cells did not exceed 5 average follicular diameters [28], while diffuse CCH was defined when it exceeded these dimensions.

Histological examination also provided a lymphocytic thyroid infiltration description. On the basis of parenchymal distribution pattern, lymphocytic infiltration was described as focal (presence of sporadic lymphocytic infiltrate centers) and diffuse (presence of multiple lymphocytic infiltrate centers in both lobes), according to the pathological diagnosis of focal or diffuse lymphocytic thyroiditis (LT).

In order to facilitate the pathologist in detailing the exact position of each TN and ensure that the histologically examined TNs correspond to those previously submitted to FNA, surgical specimens were marked with reference points to indicate the correct orientation of the thyroid, and the pathologist received the specimen along with a detailed map of the TN location as described in the US report.

### 2.6. Mutational Analysis of the RET Proto-Oncogene

Mutation analysis of the RET proto-oncogene involved DNA extraction and polymerase chain reaction (PCR) amplification. QIAamp blood kit (Qiagen, Hilden, Germany) was used to isolate high-molecular-weight DNA from peripheral blood leukocytes according to the manufacturer’s protocol. PCR amplification of nucleotide sequences of exons 8 through 16 was performed with primers designed from flanking intronic sequences according to the published sequences [29,30].

## 3. Results

### 3.1. Patients’ Characteristics of the Study Group

As shown in Table 1, of the 11 patients included in our study cohort, 4 were males (age range 42–73 years) and 7 were females (age range 37–77 years). Based on the FNA-CT cut-off, patients were subdivided into two groups: the first (6 patients) with FNA-CT higher (≥36 pg/mL) and the second (5 patients) with FNA-CT lower (<36 pg/mL) than the cut-off. All the patients were submitted to total thyroidectomy, and a detailed histological examination was available for each case; none of them showed MTC at histology or RET proto-oncogene mutations.

### 3.2. Patients in the Study Group with High FNA-CT Values

As displayed in Table 1, in this group of 6 patients, FNA-CT values ranged from 271 to >2000 ng/mL. All these patients showed diffuse CCH in the thyroid lobe submitted to FNA (see Figure 1). Furthermore, two patients in this group showed THY 2 cytology and histology-documented diffuse LT associated in one case with HN, while the other had only focal LT. Among the other 4 patients, two showed THY 3-A cytology, and histology documented in one case a single HN, while the other had MNG. Finally, the remaining two patients had THY-3B cytology, and histology resulted in one case of FA, while the other had an invasive encapsulated PTC-follicular variant (IEPTC-FV).

### 3.3. Patients in the Study Group without Increased FNA-CT Values

In this group of five patients, FNA-CT values were markedly lower, ranging from <2 to 11 pg/mL. None of these patients had diffuse CCH in the thyroid lobe submitted to FNA. Two of these patients showed THY 4 cytology and histology documented in one case, IEPTC-FV of the left lobe without CCH, while diffuse CCH was present in the right lobe; the other patient had PTC-classic variant (PTC-CV) of the left lobe associated with focal CCH (see Figure 2). Among the other 3 patients, 2 of them showed THY 3-A cytology and histology documented in one patient HN associated with diffuse LT, and the other had MNG; the remaining patient had THY 3-B cytology corresponding to a FC at histology. As also reported in the table, all three patients showed focal bilateral CCH on histology.

### 3.4. Comparison between TN Dimensions, Type of CCH, and Basal and Stimulated Serum CT in Patients of the Study Group

As previously reported, high (>36 pg/mL) FNA-CT concentrations were always associated with diffuse CCH of the lobe submitted to FNA, while absent or focal CCH were associated with FNA-CT <36 pg/mL. Furthermore, the median values of basal (32.5 pg/mL, range 26–52) and stimulated serum CT (177 pg/mL, range 146–250) found in the group of patients with high FNA-CT and diffuse CCH were higher than the basal (17 pg/mL, range 15–31) and stimulated serum CT (40 pg/mL, range 35–210) observed in the group of patients with normal FNA-CT and absent or focal CCH. Interestingly, the only patient with undetectable FNA-CT in spite of rather increased basal (31 pg/mL) and stimulated (210 pg/mL) serum CT concentrations had an IEPTC-FV, associated with a diffuse CCH only in the contralateral lobe. Finally, no correlation was found between basal (Figure 3) and stimulated serum CT concentrations and the size of the TN submitted to FNA, indicating that the CT found in the FNA-CT originated from the diffuse CCH and not from the nodule itself.

### 3.5. Patients’ Characteristics of the Control Group

As detailed in Table 2, the control group consisted of 10 patients, 9 females, and 1 male, with a median age of 62 years, who were investigated in our previous study [2]. In all these patients, histological examination confirmed the presence of MTC. In order to compare their records with the records of the present study group, we reanalyzed all the data from the control group. The median basal CT value was 550 pg/mL (range 21–2000), while the median stimulated serum CT value was 721 pg/mL (range 227–1521). In contrast with the study group, a highly significant positive correlation was observed between the sizes of TNs and basal serum CT concentrations (R² = 0.8726); see also Figure 4. Notably, most of them (8 out of 10) displayed suspicious (n. 3 of TI-RADS 4) or malignant (n. 5 of TI-RADS 5) US features, while only two had intermediate US risk (TI-RADS 3). On the other hand, 6 out of 10 had suspicious/malignant (THY 4/THY 5) cytology, only one had intermediate risk (THY 3-B) cytology, and three resulted in not being diagnostic. Finally, in almost all cases (9 out of 10), FNA-CT values exceeded 2000 pg/mL, except for one case where it measured 860 pg/mL. Unfortunately, the precise concentration of CT in FNA-CT exceeding 2000 pg/mL remains unknown since dilutions were not performed.

## 4. Discussion

FNA-CT is currently reported as a useful tool for the diagnosis of both primary and lymph-node metastatic MTC [1,3]. However, as preliminary reported by Diazzi et al. [14], elevated FNA-CT values may be detected in TNs other than in MTC when associated with CCH due to contamination of the needle through the aspiration route within a CCH area. The values reported in this series presented a significant degree of overlap with FNA-CT values obtained from MTC, not allowing for differentiation of these conditions. Despite the evidence of false-positive FNA-CT due to CCH, all patients in this study underwent a total thyroidectomy. Although histological examination in several cases showed a benign thyroid lesion associated with CCH, the clinical relevance of this observation remained undefined, as the study suggested the need for surgical intervention in the presence of CCH. After the publication of our study on the usefulness of FNA-CT in MTC diagnosis [2], we continued to use this technique in patients with TNs and elevated serum CT. During the review process of our clinical records, we focused our attention on the present group of 11 patients with TNs and elevated serum CT who were also submitted to FNA-CT and underwent total thyroidectomy independently from FNA-CT results. None of them showed MTC at histology, either the 6 patients with elevated FNA-CT or the 5 cases without increased FNA-CT. The results of this study group were compared with all the records of our original control group of 10 patients who presented MTC at histology [2]. We observed a direct correlation between the sizes of the MTC, basal, and stimulated CT values in the control group, while in contrast, there was no correlation between TN size, basal, or stimulated CT values in the study group without MTC. However, in the study group, a clear correlation between basal, stimulated CT, and FNA-CT values was associated with the pattern of diffuse CCH at histology rather than the TN size, while the median values of basal and stimulated serum CT found in the control group of MTC patients were higher than the corresponding values observed in the study group of patients with CCH.

Interestingly, the comparison between FNA-CT in the study group and the control group showed FNA-CT concentrations were clearly higher in patients with MTC when compared to those without MTC. Although in MTC patients we were unable to calculate the precise CT concentrations of FNA-CT, values above 2000 pg/mL were found in 9 out of 10 MTC patients and only in one patient in the diffuse CCH group. Furthermore, among patients in the study group, diffuse CCH was always associated with elevated (i.e., above the pre-established cut-off of 36 pg/mL) FNA-CT, while focal CCH was associated with FNA-CT below 36 pg/mL, suggesting that the probability of having needle contamination was correlated to the number of C-cells located near TNs submitted to aspiration biopsy. Moreover, higher values of serum basal and stimulated CT were found in patients with diffuse CCH than in those with focal CCH, indicating that the circulating CT actually originated from the thyroid C-cells and not from an ectopic source.

The review of the other clinical records of patients in the study group with high FNA-CT and diffuse CCH showed that five of them presented with TN without ultrasound and cytologically suspicious features, and they all received a benign diagnosis at histology. In contrast, the re-analysis of the records of the MTC group showed most of them (8 out of 10) had suspicious or malignant US features, and 6 out of 10 also had suspicious/malignant cytology. Considering these results, the patients in the study group with elevated FNA-CT probably did not need surgery, which was only based on falsely positive FNA-CT values. On the other hand, the appropriate surgical recommendation for the remaining 6 patients in our series was confirmed by the presence of malignancy at histology in 4 of them (1 with high FNA-CT and 3 with normal FNA-CT), while the remaining 2 cases (with normal FNA-CT) had benign histology. According to the results of this study, we believe that the FNA-CT cut-off value of 36 pg/mL, originally considered diagnostic for MTC in our previous study [2], should be revised and updated. In our series, all TNs with homolateral diffuse CCH had FNA-CT concentrations largely above 36 pg/mL, reaching, in one case, a value of >2000 pg/mL. On the other hand, although 9/10 patients with MTC had FNA-CT concentrations >2000 pg/mL, one had a lower value (860 pg/mL), clearly overlapping with the FNA-CT concentrations observed in the present series of patients with TNs associated with diffuse CCH. Although our available data strongly suggest that FNA-CT concentrations are higher in FNA from MTC than in CCH, further studies, carried out with appropriate dilutions to calculate the precise CT concentration in the aspirate fluid, are needed to see if a clinically valid cut-off of FNA-CT concentration could be proposed to differentiate MTC from CCH.

Thus, the suggestion of Diazzi et al. [14] that surgery is mandatory for FNA-CT > 1000 pg/mL and strongly recommended for values between 36 and 1000 pg/mL for the risk of MTC or neoplastic CCH should be subject to reconsideration. In addition, the analysis of basal CT and stimulated CT values with the size of the TN submitted to FNA, which were significantly higher in MTC patients than in the diffuse CCH group, could represent another valuable element to determine the presurgical risk of MTC and the propensity towards surgery. The results of basal, stimulated, and FNA-CT, along with US and cytological features of TNs, could probably be used in a combined way in an algorithm to estimate the risk of MTC and the consequent surgical indication. This aspect could be particularly pertinent in older patients exhibiting US and cytological features that are not suspicious, along with slightly elevated levels of serum basal CT and the coexistence of HT or MNG, who may only need follow-up. The development and validation of such an algorithm are beyond the purpose of the present report and need further study. A further difficulty in the pre-surgical differentiation between MTC and CCH is that the biological behavior of CCH is different in different clinical contexts. Most of the CCH associated with MNG and Hashimoto’s thyroiditis, believed to represent “reactive” or “physiologic” (non-neoplastic) CCH, is different from (pre)neoplastic CCH, typically found in familial MTC/MEN-2 syndromes [16], where surgery may be mandatory even before definitive malignant transformation in patients carrying “high-risk” genomic RET mutations.

Another noteworthy consideration, given the real risk of false-positive results attributed to diffuse CCH, involves ensuring precise patient selection for FNA-CT. This selection process should encompass not only basal CT levels but also ultrasound features, cytological characteristics, familial history, and RET status assessment.

Finally, recent studies provided evidence that genomic analysis of cytological specimens by an RNA-sequencing MTC classifier may allow the preoperative identification of MTC in unselected TNs submitted to FNA with very high sensitivity and specificity [31,32]. The diffusion of this very promising approach is presently limited by its high cost, and, to our knowledge, no direct comparison between the genomic and biochemical (CT) characterization of cytological specimens has been reported so far.

## 5. Conclusions

In conclusion, the reanalysis of our series of MTC and the comparison with a highly selected series of patients with diffuse CCH without MTC underline the importance of a rational clinical approach in patients with TNs with increased serum CT and high FNA-CT concentrations higher than the conventionally proposed cut-offs. The evaluation of basal CT and stimulated CT values compared to the size of the TN submitted to FNA should always support the FNA-CT results, together with the US pattern and the cytological features of the TN. Further studies carried out on larger patient numbers are needed to verify the possibility of including FNA-CT in a more complex algorithm to estimate the presurgical risk of MTC and the consequent surgical indication. In any case, it is evident from the present study that the use of low FNA-CT cut-off values, such as those proposed so far, should be used cautiously in patients with TNs and increased serum CT due to unavoidable false-positive FNA-CT values caused by the presence of diffuse CCH in the lobe submitted to aspiration. Finally, this problem could be definitively circumvented in the near future by genomic analysis of FNA, a very promising tool for the preoperative diagnosis of MTC.

## Figures and Tables

**Figure 1 cancers-16-00210-f001:**
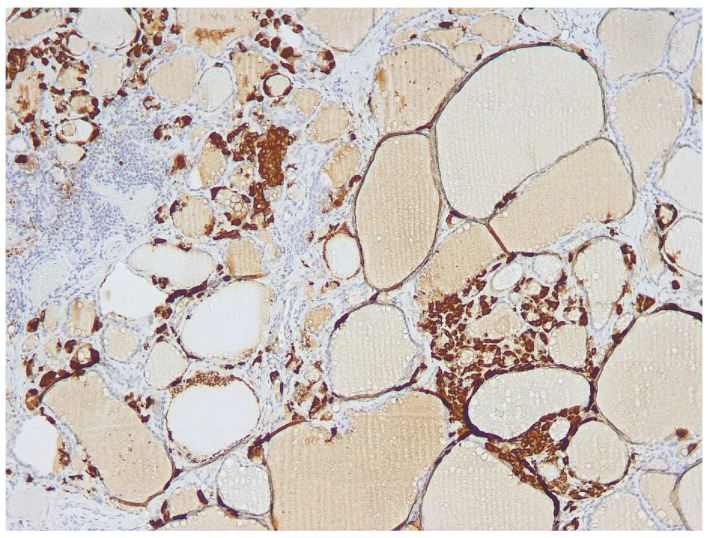
Diffuse C-cell hyperplasia (cells stained brown at immunohistochemistry) of the thyroid lobe was submitted to FNA.

**Figure 2 cancers-16-00210-f002:**
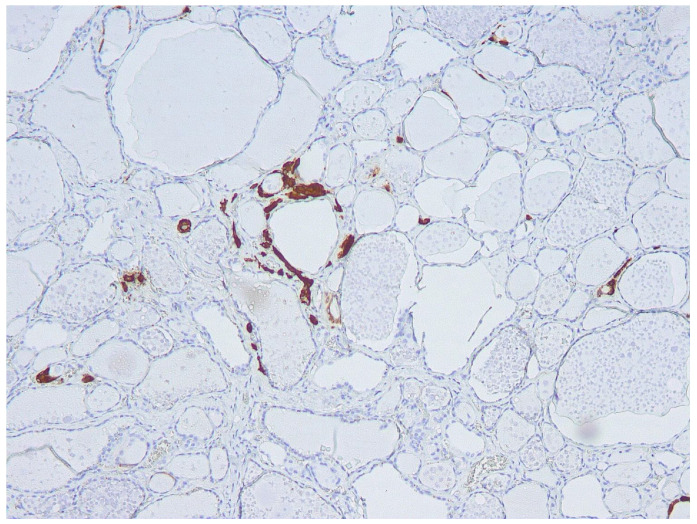
Focal C-cell hyperplasia (cells stained brown at immunohistochemistry) in the thyroid lobe was submitted to FNA.

**Figure 3 cancers-16-00210-f003:**
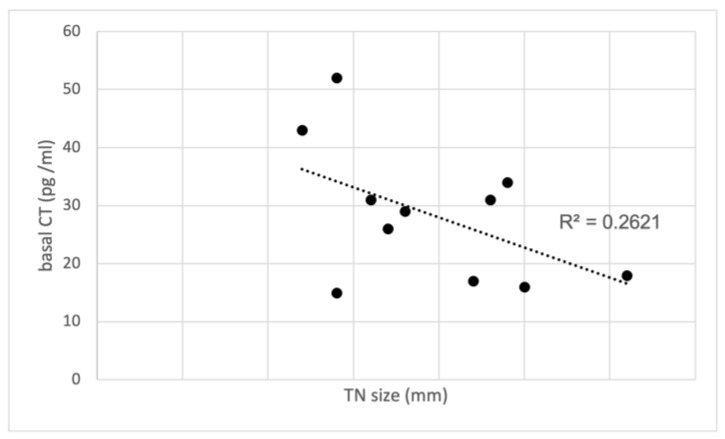
Correlation between basal CT and TN size in the study group with CCH. CT, calcitonin; TN, thyroid nodules.

**Figure 4 cancers-16-00210-f004:**
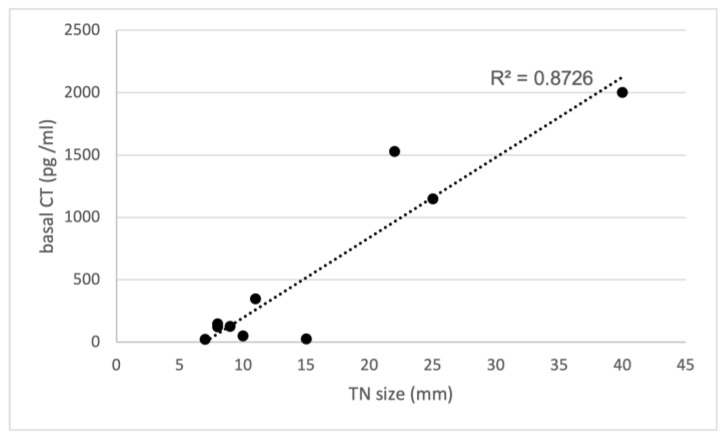
Correlation between basal CT and TN size in the control group with MTC. CT, calcitonin; TN, thyroid nodules; R² = 0.8726.

**Table 1 cancers-16-00210-t001:** Patients’ characteristics, biochemical, ultrasound, and pathological results of the study group.

Age Sex	Lobe/Size/TI-RADS Score	Cytology	Nodule Histology	Basal CT (pg/mL)	FNA-CT (pg/mL)	Stimulated CT (pg/mL)	C-Cell Histology
69/F	Lobe sx/12 mm/TI-RADS 3	THY 2	Focal LT	43	>2000	250	Diffuse bilateral CCH
74/F	Lobe dx/18 mm/TI-RADS 3	THY 3-A	MNG	29	1480	165	Diffuse CCH right lobe
60/F	Lobe sx/24 mm/TI-RADS 3	THY 2	HN in LT	34	1100	180	Diffuse bilateral CCH
42/M	Lobe sx/17 mm/TI-RADS 3	THY 3-B	FA	26	660	146	Diffuse CCH left lobe
50/F	Lobe dx/14 mm/TI-RADS 5	THY 3-B	PTC-FV	52	280	243	Diffuse bilateral CCH
51/M	Lobe dx/23 mm/TI-RADS 3	THY 3-A	HN	31	271	174	Diffuse bilateral CCH
77/F	Isthmus/25 mm/TI-RADS 4	THY 3-A	MNG	16	11	44	Focal bilateral CCH
73/M	Lobe dx/31 mm/TI-RADS 3	THY 3-A	HN in LT	18	10	40	Focal bilateral CCH
37/F	Isthmus/22 mm/TI-RADS 4	THY 3-B	FC	17	8	35	Focal bilateral CCH
50/F	Lobe dx/14 mm/TI-RADS 5	THY 4	PTC-CV	15	<2	39	Focal CCH left lobe
49/M	lobe sx/16 mm/TI-RADS 4	THY 4	IEPTC-FV	31	<2	210	Diffuse CCH right lobe

TI-RADS and THY-cytology, see text; LT, lymphocytic thyroiditis; MNG, multinodular goiter; HN, Hyperplastic nodule; FA, Follicular Adenoma; PTC-FV, Papillary thyroid carcinoma (PTC)-follicular variant; FC, Follicular Carcinoma; PTC-CV, PTC-classic variant; CT, calcitonin; FNA-CT, CT in fine needle aspiration; CCH, C-cell hyperplasia.

**Table 2 cancers-16-00210-t002:** Characteristics of the control group: biochemical, ultrasound, and pathological results.

Age Sex	Lobe/Size/TI-RADS Score	Cytology	Basal CT (pg/mL)	FNA-CT (pg/mL)	Stimulated CT (pg/mL)	Histology
77/F	Lobe dx/7 mm/TI-RADS 5	ND	21	>2000	670	MTC
68/F	Lobe sx/8 mm/TI-RADS 5	ND	120	>2000	420	MTC
64/F	Lobe dx/8 mm/TI-RADS 5	THY 4	145	2000	227	MTC
60/F	Isthmus/9 mm /TI-RADS 4	THY 5	123	>2000	524	MTC
50/F	Lobe dx/10 mm/TI-RADS 4	ND	47	>2000	1319	MTC
55/F	Lobe dx/11 mm/TI-RADS 5	THY 5	345	>2000	1521	MTC
57/M	Lobe sx/15 mm/TI-RADS 5	THY 5	23	>2000	369	MTC
60/F	Isthmus/22 mm /TI-RADS 4	THY 3-B	1526	860	NP	MTC
53/F	Lobe dx/25 mm/TI-RADS 3	THY 4	1146	>2000	NP	MTC
76/F	Lobe sx/40 mm/TI-RADS 3	THY 4	2000	>2000	NP	MTC

TI-RADS and THY-cytology, see text; CT, calcitonin; FNA-CT, CT in fine needle aspiration; ND, not diagnostic; NP, not performed; MTC, Medullary thyroid cancer.

## Data Availability

All data generated or analyzed during this study are included in this published article.

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
