# Peer review of "Diffuse C-Cells Hyperplasia Is the Source of False Positive Calcitonin Measurement in FNA Washout Fluids of Thyroid Nodules: A Rational Clinical Approach to Avoiding Unnecessary Surgery"

_cancers, 2024, doi:10.3390/cancers16010210_

Round 1
Reviewer 1 Report
Comments and Suggestions for Authors
The paper addresses a very interesting and clinically important problem related to the diagnosis of patients with abnormal calcitonin levels in the blood. The authors analyze the cases of patients who had elevated calcitonin values in their blood, but ultimately did not have medullary carcinoma. In this group, a subgroup of 6 patients with a high calcitonin value in the washout fluid of FNA (FNA-CT) of thyroid nodules is particularly interesting. The observation that all these patients had features of diffuse C-cell hyperplasia in the biopsied lobe is very interesting. The results are presented in a clear way, but the discussion leaves the reader unsatisfied considering the authors' extensive experience in the field.
In particular, the conclusion formulated by the authors is of a very general nature. Obviously, in every clinical procedure, a "rational clinical approach... through the evaluation of all clinical parameters (familial, ultrasound, cytological and biochemical)" is indicated when making a decision about surgical treatment. It is a pity that the authors, based on their many years of experience, did not attempt to formulate more practical recommendations. Perhaps this would be possible by the comparison of the indicated significant clinical parameters between the group of patients without the presence of medullary carcinoma and the group of patients with high FNA-CT in whom the presence of medullary carcinoma was finally confirmed.
In addition, the manuscript raises the following questions:
1. The authors declared that the study was retrospective, but they also noted that "Cytological examination was made by an experienced thyroid pathologist (M.L.L.), who was unaware of FNA-CT results." It is puzzling whether this means that as part of routine practice, the result of the FNA-CT test was concealed from the pathologist, or whether for the purposes of this work the slides were evaluated by a re-selected pathologist unaware of the examination in which he participated?
2. Was the presence of factors influencing the reliability of CT assessment in the blood of the examined patients (e.g. medications used, coexistence of clinically and laboratory-confirmed Hashimoto's disease) analyzed? There is no information about this in the text of the paper.
Author Response
Please, see the attached file

Reviewer 2 Report
Comments and Suggestions for Authors
The present work is an excellent analysis of the histological outcome of surgery in 11 patients with elevated serum calcitonin, but benign FNA findings, classified by the concentration of their calcitonin in the FNA material. The results are impressive, in that they suggest that usually, the presence of CCH (focal or diffuse) could masque itself biochemically as MTC, even though it ends up to be benign.
1. Most sentences have linguistic issues, requiring corrections. Please have the entire manuscript revised by a native English speaker.
Examples of errors (in brief):
Line 52 "can be interfered" requires rephrasing.
Lines 53-54 "particularly in presence" needs to be rephrased to "particularly in the presence"
Line 54 "For these limitations" needs to be rephrased to "because of these limitations"
Line 54 "the routinely use" needs to be rephrased to "the routine use"
Line 55 "has not yet generally proposed by the current guidelines" needs to be rephrased to " has not yet been generally proposed by the current guidelines"
and so on...
2. Calcium stimulation reference values for calcitonin are unclear. You consider borderline values to range between 30-100 pg/mL, while positive values are those >130 pg/ml in males and >95 pg/ml in females. What happens with values 100-130pg/ml in males and 95-130pg/ml in females?
3. Why would you recommend thyroid surgery for large MNG? Please note that most benign nodules by FNA are actually benign...
4. Please cite and incorporate in the discussion on prevention of unnecessary thyroidectomies the following citation:
https://pubmed.ncbi.nlm.nih.gov/35436327/
5. Please add the percentages of C-cells required to establish the diagnosis of diffuse and focal CCH on Histology (section 2.5).
6. When you performed an FNA of the thyroid nodules, in which you measured the washout calcitonin, how did you confirm that these nodules were the ones examined histologically and reported as malignant or benign? Please explain in the methods section.
7. Given the potentially benign nature of PTC-FV, please comment on the features of these tumors identified by histology.
7. The study is limited by the small number of cases analyzed and the single institution experience. These limitations should be highlighted in the text.
Comments on the Quality of English Language
Needs quite some editing by a native English speaker.
Author Response
Please, see the attached file

Round 2
Reviewer 1 Report
Comments and Suggestions for Authors
-